# Evaluating the Logical Reasoning Abilities of Large Reasoning Models

## Abstract

Large reasoning models, which are post-trained on long chain-of-thought (long CoT) data with reinforcement learning, achieve state-of-the-art performance on mathematical, coding, and domain-specific reasoning benchmarks. However, their logical reasoning capabilities—fundamental to human cognition and independent of domain knowledge—remain understudied. To address this gap, we introduce **LogiEval**, a holistic benchmark for evaluating logical reasoning in large reasoning models. LogiEval spans diverse reasoning types (deductive, inductive, analogical, and abductive) and task formats (e.g., logical sequence, argument analysis), sourced from high-quality human examinations (e.g., LSAT, GMAT). Our experiments demonstrate that modern reasoning models excel at 4-choice argument analysis problems and analogical reasoning, surpassing human performance, yet exhibit uneven capabilities across reasoning types and formats, highlighting limitations in their generalization. Our analysis reveals that human performance does not mirror model failure distributions. To foster further research, we curate **LogiEval-Hard**, a challenging subset identified through a novel screening paradigm where small-model failures (Qwen3-30B-A3B) reliably predict difficulties for larger models. Modern models show striking, consistent failures on LogiEval-Hard. This demonstrates that fundamental reasoning bottlenecks persist across model scales, and establishes LogiEval-Hard as both a diagnostic tool and a rigorous testbed for advancing logical reasoning in LLMs.

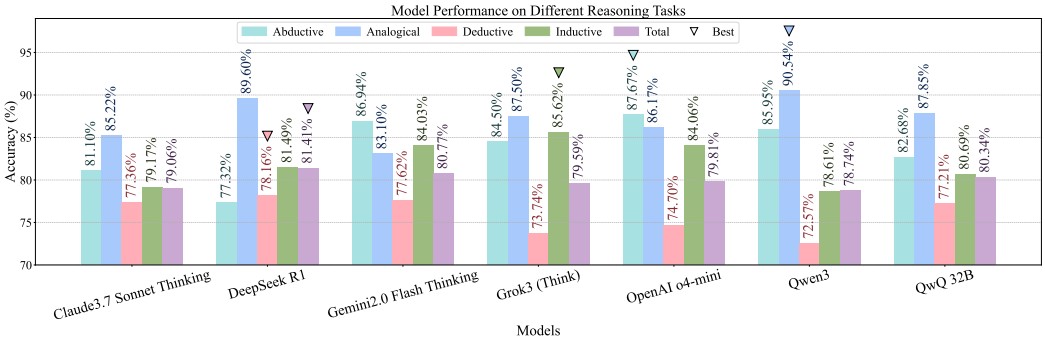

Figure 1: Model performance comparison on different subtasks

## 1 Introduction

Large language models (LLMs) with advanced reasoning capabilities—often termed reasoning language models or large reasoning models—have become pivotal in both industry (Guo et al., 2025; Qwen Team, 2025a; Anthropic, 2025; xAI, 2025; OpenAI, 2025) and academia (Muennighoff et al., 2025). These models acquire their "thinking" ability through post-training (Kumar et al., 2025) on long chain-of-thought (long CoT) examples (Chen et al., 2025a), which are either human-annotated (Muennighoff et al., 2025) or generated via reinforcement learning (RL) over action spaces (Xu et al., 2025). Such data enables LLMs to perform multi-step reasoning and plan for complex tasks.

This approach, now widely adopted (Qwen Team, 2025b;a), enables models to generate intermediate reasoning steps—either implicitly or explicitly—before delivering final outputs. Despite their success, evaluation remains skewed toward domain-specific benchmarks: mathematical reasoning (Ye et al., 2025; Lightman et al., 2023), coding (Penedo et al., 2025; Jimenez et al., 2024), and knowledge-intensive tasks (Wang et al., 2024; Rein et al., 2024). A critical gap persists in assessing logical reasoning—a fundamental capability independent of domain knowledge—leaving the true generalization of these models unclear.

We aim to give a systematic empirical evaluation of reasoning language models on logical reasoning, covering inductive (Sinha et al., 2019), deductive (Saparov & He, 2023; Tian et al., 2021; Sanyal et al., 2022; Parmar et al., 2024), abductive (Del & Fishel, 2023; Nguyen et al., 2023), and analogical (Petersen & van der Plas, 2023; Qin et al., 2024; Wijesiriwardene et al., 2023) reasoning types – the four basic logical reasoning categories (Liu et al., 2025). To this end, several benchmark datasets are available in exam-originated (e.g., LSAT ,Civil Service tests) (Liu et al., 2021; Yu et al., 2020) or synthetically-generated (Sinha et al., 2019; Saparov & He, 2023) forms. However, existing benchmarks typically focus on one aspect of reasoning such as inductive (Sinha et al., 2019) or deductive (Saparov & He, 2023; Tian et al., 2021; Sanyal et al., 2022; Parmar et al., 2024) reasoning. In addition, these challenges are typically represented by a single problem format, mostly multi-choice question answering (Liu et al., 2021; Yu et al., 2020), which may be subject to data artifacts (Ye et al., 2024; Chen et al., 2025b). Finally, existing benchmark suites such as GLoRE (liu et al., 2023) and LogiTorch (Helwe et al., 2022) assemble multiple reasoning benchmarks for increased reliability, yet they lack a unified representation, and recent LLMs have achieved very high performances on these datasets (liu et al., 2025).

To address this gap, we introduce **LogiEval**, a comprehensive logical reasoning benchmark curated from diverse human examinations (e.g., LSAT, GMAT, Civil Service Exams), covering multiple reasoning types (deductive, inductive, analogical and abductive) and tasks (e.g., essential part, artificial language, syllogism) and problem formats(muti-choice QA, 3-way classification), providing a unified evaluation suite for assessing fundamental reasoning abilities. The benchmark consists of 6,235 instances in total. Some example questions are shown in Figure 2. We evaluate state-of-the-art models—including DeepSeek-R1 (Guo et al., 2025), Qwen3 (Qwen Team, 2025a), Claude-3.7-Sonnet (Anthropic, 2025), Grok-3 (xAI, 2025), Gemini-2.0-Flash-thinking, and OpenAI o4-mini (OpenAI, 2025)—revealing two key findings: (1) reasoning LLMs give varying performances across various tasks — while the *essential part* task is easily solved by most models, *artificial language* and *syllogism* tasks remain rather challenging. (2) Different reasoning LLMs excel on different tasks and reasoning types, without a winner across the board. (3) LogiEval is challenging to existing reasoning LLMs, with the best-performing model giving around 80% overall accuracy. (4) A subset of the most challenging questions to one model (i.e., QwQ-32B) is also the most challenging to all the other models, showing that there is some common challenge to existing reasoning LLMs. Accordingly, we extract the most difficult subset of LogiEval and name it **LogiEval-Hard** to further resolve the saturation issue. LogiEval is gated behind manual review to prevent adversarial optimization. We release our dataset on HuggingFace and enable access control to avoid misuse. Researchers who apply for access should adhere to usage guidelines.

Overall, our contributions are: (1) We introduce LogiEval, a comprehensive logical reasoning benchmark curated from 7 high-stakes human examinations that unifies four fundamental reasoning types across 10 task formats - the first to combine diverse tasks and question formats into one evaluation suite. (2) Through systematic evaluation of 7 cutting-edge reasoning models (2025 releases), we reveal critical gaps in LLM capabilities: while models exceed human performance on 4-choice argument analysis, they show catastrophic failures on syllogisms and exhibit inverse difficulty correlations, solving "hard" human problems while failing "medium" ones. (3) We develop a novel screening paradigm where small-model failures (Qwen3-30B-A3B) reliably predict universal challenges, producing LogiEval-Hard - the first benchmark subset where all models show striking, consistent failures (avg 37.97% accuracy), exposing fundamental reasoning bottlenecks that persist across model scales.

### Deductive

Context: All birds can fly. Penguins are birds.
Question:
Conclusion I: Penguins can fly;
Conclusion II: Some birds can swim.
**Which conclusion follows?**
A. Only conclusion I follows  √
B. Only conclusion II follows
C. Either I or II follows
D. Neither I nor II follows
E. Both I and II follow

### Inductive

Context: For the past two months stereo shops all over the city have been hit by burglars in the early morning hours. Sergeant Adams tells Officer Bryant that he should carefully watch the stores in his area that specialize in stereo equipment.
Question: **Which one of the following situations should Officer Bryant investigate?**
A. A truck with its motor running backed up to the rear door of the House of Stereos at 2 a.m.  √
B. An elderly couple window shopping at the House of Stereos at 10 p.m.
C. a delivery van marked House of Stereos parked in the rear of the store at 11:30 p.m.
D. Two teenaged boys intently examining a stereo system in the window of House of Stereos at midnight

### Analogical

Context: Identify the relationship between the first pair of words and select the answer that best replicates the same relationship in the second pair.
Question: **Cup is to coffee as bowl is to**
A. dish
B. soup  √
C. spoon
D. food

### Abductive

Context: A research study revealed that, in most cases, once existing highways near urban areas are widened and extended in an attempt to reduce traffic congestion and resulting delays for motorists, these problems actually increase rather than decrease.
Question: **Which one of the following, if true, most helps to explain the discrepancy between the intended results of the highway improvements and the results revealed in the study?**
A. Widened and extended roads tend to attract many more motorists than used them before their improvement  √
B. Typically, road widening or extension projects are undertaken only after the population near the road in question has increased and then leveled off, leaving a higher average population level
C. As a general rule, the greater the number of lanes on a given length of highway, the lower the rate of accidents per 100,000 vehicles traveling on it
D. Rural, as compared to urban, traffic usually includes a larger proportion of trucks and vehicles used by farmers

Figure 2: Examples from the LoigEval benchmark

## 2 LOGIEVAL

Our motivation for the benchmark is to cover all the question types from all sorts of examinations concerning logical reasoning. Other than testing only one unique aspect of logical reasoning with one benchmark, our intention is to give the LLM providers the freedom to get an integrated solution with our benchmark. Moreover, compared to the benchmarks that are commonly used for testing new models that require intensive domain knowledge, a key advantage of LogiEval is its domain-agnostic nature. By decoupling reasoning from specialized knowledge, it mirrors the fundamental cognitive abilities shared across humans—regardless of educational background—and provides a purer measure of a model's reasoning capacity. This makes LogiEval particularly valuable for evaluating whether LLMs can generalize logical principles beyond pattern recognition in narrow domains.

### 2.1 DATA COLLECTION AND CURATION

LogiEval is constructed from carefully selected logical reasoning sections of high-stakes human examinations, including the Chinese Civil Service Examination, Law School Admission Test (LSAT), Graduate Management Admission Test (GMAT), Banking Personnel Selection (IBPS), Common Admission Test (CAT), and several standardized IQ and aptitude tests. These examinations were chosen because they each contain dedicated logical reasoning components that have been rigorously developed to assess human reasoning capabilities. All questions were sourced from publicly available practice materials and are used strictly for academic research purposes. We maintain the original language of each examination (either English or Chinese) to preserve the linguistic nuances of logical reasoning tasks, avoiding potential biases introduced by translation (Liu et al., 2023; Song et al., 2025). This approach makes LogiEval a genuinely bilingual benchmark that can evaluate reasoning abilities across languages.

The difficulty level of the benchmark faithfully reflects that of the original examinations, ensuring that it maintains the same discriminative power for evaluating reasoning capabilities as the tests demonstrate for human test-takers. By preserving both the content and difficulty characteristics of these established examinations, LogiEval provides an authentic assessment environment for evaluating large language models' logical reasoning abilities.

After de-duplication and validity verification, we obtain 6,235 high-quality problems, each annotated with gold labels. In addition to these labels, our dataset includes supplementary metadata such

Table 1: The dataset statistics of LogiEval

| Reasoning Type | Count | Task Format | Count | Task Format | Count | Options | Count |
|---|---|---|---|---|---|---|---|
| abductive | 961 | argument analysis | 1,354 | logical sequence | 135 | 2 | 10 |
| analogical | 379 | artificial language | 195 | odd one out | 71 | 3 | 3,766 |
| deductive | 3,681 | blood relations | 91 | situational judgement | 310 | 4 | 1,575 |
| inductive | 1,214 | definition matching | 324 | syllogism | 308 | 5 | 879 |
| Total | 6,235 | essential part | 55 | textual entailment | 3,402 | 6 | 5 |

as difficulty level, human accuracy rate, and explanations, derived directly from post-exam statistics. To enable fine-grained analysis, we further annotate each question with its task format and reasoning type. Annotation details are provided in Appendix A

Task formats are categorized into 10 distinct types based on question structure: *logical sequence, essential part, artificial language (coding and decoding), blood relation, situation judgment, syllogism, definition matching, argument analysis, odd one out, and textual entailment*. To our knowledge, we are the first to cover *artificial language* questions, *essential part* questions, and *odd one out* questions in logical reasoning. For reasoning types, we employ a hybrid annotation pipeline: Qwen3-30B-A3B first proposes one of four reasoning categories (analogical, deductive, inductive, or abductive), followed by human verification to mitigate potential model hallucinations. Three annotators independently review each label, with final assignments determined by majority vote. Figure 2 shows 4 examples from LogiEval representing each reasoning type.

The dataset is partitioned into a few-shot development set and a test set. The development set includes 5 representative examples per task format, accompanied by task-specific instructions to guide model adaptation. The test set comprises 6,174 problems, while the development set contains 65, ensuring robust evaluation across diverse reasoning scenarios.

## 2.2 DATA STATISTICS

As shown in Table 1, LogiEval comprises 6,235 instances distributed across four reasoning types and ten task formats. Deductive reasoning constitutes the largest category with 3,681 instances, followed by inductive (1,214), abductive (961), and analogical reasoning (379). The task format distribution reveals textual entailment as the most prevalent (3,402 instances), followed by argument analysis (1,354), while niche formats like essential part (55) and odd one out (71) represent specialized challenges. The benchmark features diverse answer options ranging from 2 to 6 choices, with 3-option questions dominating (3,766), followed by 4-option (1,575) and 5-option formats (883). This composition ensures comprehensive evaluation across reasoning paradigms while maintaining examination authenticity through varied question structures.

## 3 EXPERIMENTS AND RESULTS

### 3.1 EXPERIMENTAL SETUP

We evaluate state-of-the-art large reasoning models released in 2025, all featuring advanced reasoning capabilities. Despite differences in scale (32B to 671B parameters), architecture (dense vs. MoE), and training strategies (RL-based long CoT vs. hybrid thinking modes), these models rank among the top 50 in the LMSYS Chatbot Arena Leaderboard[1] as of May 2025.

For consistency, we convert each instance into minimal-design prompts and extract answers using regex-based pattern matching (details in Appendix B). Accuracy is computed against gold labels, with task-specific normalization for multi-format evaluation.

For consistent evaluation, each data instance is converted into a standardized, minimal-design prompt. To extract answers from model responses, we apply exact string matching, following the approach of Er & Cicekli (2013). The extracted answers are then compared against the gold labels to compute accuracy. For model evaluation, we use the official API by the LLM provider. Apart

---

[1] https://huggingface.co/spaces/lmsys/chatbot-arena-leaderboard

Table 2: The performance of large reasoning models on different tasks of LogiEval

| Task Format | CLAUDE3.7 SONNET THINKING | DEEPSEEK R1 | GEMINI2.0 FLASH THINKING | GROK3 (THINK) | OPENAI O4-MINI | QWEN3 -235B A22B | QWQ 32B |
|---|---|---|---|---|---|---|---|
| Argument Analysis | 85.70% | 81.20% | 87.01% | 88.22% | **89.90%** | 87.72% | 88.19% |
| Artificial Language | 64.30% | 72.34% | 80.93% | **91.67%** | 82.71% | 80.99% | 54.52% |
| Blood Relations | 59.60% | **74.94%** | 73.79% | 73.68% | 72.04% | 58.24% | 71.62% |
| Definition Matching | 90.91% | 91.65% | 91.23% | 88.60% | 87.35% | **96.12%** | 94.71% |
| Essential Part | **100%** | **100%** | 95.99% | 95.30% | **100%** | **100%** | 94.88% |
| Logical Sequence | 86.04% | 89.24% | 91.45% | 93.68% | **95.18%** | 83.46% | 92.44% |
| Odd One Out | 77.93% | 79.05% | 81.28% | **88.69%** | 85.57% | 83.99% | 78.26% |
| Situational Judgement | **77.34%** | 74.43% | 74.26% | 73.87% | 69.14% | 72.95% | 67.57% |
| Syllogism | 70.61% | 73.38% | 70.19% | 51.86% | 54.85% | 56.93% | **73.46%** |
| Textual Entailment | 79.55% | **83.77%** | 75.20% | 77.81% | 78.48% | 82.72% | 77.50% |
| Total | 79.06% | **81.41%** | 80.77% | 79.59% | 79.81% | 78.74% | 80.34% |

from that, we host a Qwen3-30B-A3B model on a server with 4 Nvidia 80G VRAM H100 GPUs for extended experiments.

**Open-weighted models** Open-weighted models are those that have released their model check-points to the public. Users can deploy their reasoning models and have access to the thinking process. We chose the following reasoning models:

DEEPSEEK R1 is released in January 2025 by DeepSeek AI. It is trained on top of DeepSeek V3 with 671 B MoE parameters. The key innovation is the massive implementation of reinforcement learning for long CoT reasoning.

QWQ 32B is a 32B parameter model developed by the Qwen team. As their first reasoning model, QwQ 32B has garnered a lot of attention for its superior performance on various reasoning tasks despite its size.

QWEN3-235B-A22B is the latest flagship reasoning model of Qwen. Released in April, 2025, It is a MoE model with 235B total parameters and 22B activated parameters. One of the key features of this model is the hybrid thinking modes, which allow users to control how much "thinking" the model performs based on the task.

**Proprietary models** Proprietary models are less open compared to open-weighted models, we can access to the responses of these models either through a ChatUI or API. We chose the following models, which are claimed to be reasoning models or have a thinking mode:

GEMINI2.0 FLASH THINKING is a model developed by Google. It was released in 2025 with 32B parameters as the company's most capable reasoning model.

CLAUDE3.7 SONNET THINKING is a model developed by Anthropic. The parameter size of this model has not been revealed. It is the company's most recent release to date.

GROK3 (THINK) is developed by xAI as its most advanced reasoning model yet. The thinking model is optimized for test-time compute and reasoning.

OPENAI O4-MINI is OpenAI's most recent release of its reasoning models. The o-series of models is trained to think for longer before responding. However, the original thinking process of these models is not observable to users. Along with o3, OpenAI claims they are the smartest models they've released to date. As no API has been provided by OpenAI for o3, we include o4-mini in our experiment.

**Human passing rate** Human performance is benchmarked using historical passing rates from source examinations. These rates reflect real-world test-taker performance, providing a robust reference for model comparison.

## 3.2 RESULTS

As shown in Table 2, our comprehensive evaluation reveals several critical insights into the logical reasoning capabilities of state-of-the-art models. Models consistently outperformed human test-takers on 4-choice argument analysis problems, with human accuracy at 85.2% compared to model performance ranging from 81.20% to 89.90%. Proprietary models like OpenAI o4-mini (89.90%) and Grok3-Think (88.22%) led in this category, aligning with prior observations of LLM overfitting to multiple-choice formats. The saturation effect—where all models cluster above 81% accuracy—suggests diminishing returns in using this format to distinguish reasoning capabilities.

Performance varied dramatically across different task formats, exposing fundamental gaps in reasoning skills. Structured deductive reasoning, such as syllogisms, proved particularly challenging, with Grok3-Think (51.86%) and OpenAI o4-mini (54.85%) performing near-random on 5-option questions, while DeepSeek-R1 achieved 73.38%, likely due to its RL-based training on formal verification tasks. Contextual analogical reasoning, like artificial language tasks, showed the highest variance (54.52%–91.67%), with Grok3-Think outperforming others by over 10 percentage points, suggesting specialized training on coding/decoding tasks. Resource-intensive formats like textual entailment (75.20%–83.77%) revealed clear scaling effects, with larger models like DeepSeek-R1 (83.77%) outperforming smaller ones like QwQ-32B (77.50%).

Notably, all models achieved 95%+ accuracy on essential part identification, with four models reaching 100%—surpassing human performance (92.3% historical average). This suggests either an inherent strength in component-based reasoning or that these tasks rely on predictable pattern recognition rather than genuine reasoning.

Error analysis revealed systematic failure patterns, with 18.3% of problems incorrectly answered by all models. Errors concentrated in abductive reasoning (32% of hard subset) and situational judgment tasks (27%), confirming that aggregate metrics mask critical reasoning deficiencies. LogiEval-Hard, our challenging subset, provides a targeted evaluation suite for these gaps, with baseline accuracies below 40% for all evaluated models.

These findings demonstrate that while modern reasoning models achieve strong examination performance through format-specific optimization, their logical reasoning capabilities remain uneven and task-dependent. LogiEval-Hard serves as a critical complement to existing benchmarks by focusing on persistent failure modes.

Table 3: The performance of large reasoning models on different reasoning types of LogiEval

| Reasoing Type | CLAUDE3.7 SONNET THINKING | DEEPSEEK R1 | GEMINI2.0 FLASH THINKING | GROK3 (THINK) | OPENAI O4-MINI | QWEN3 -235B A22B | QWQ 32B |
|---|---|---|---|---|---|---|---|
| Abductive | 81.10% | 77.32% | 86.94% | 84.50% | **87.67%** | 85.95% | 82.68% |
| Analogical | 85.22% | 89.60% | 83.10% | 87.50% | 86.17% | **90.54%** | 87.85% |
| Deductive | 77.36% | **78.16%** | 77.62% | 73.74% | 74.70% | 72.57% | 77.21% |
| Inductive | 79.17% | 81.49% | 84.03% | **85.62%** | 84.06% | 78.61% | 80.69% |
| Total | 79.06% | **81.41%** | 80.77% | 79.59% | 79.81% | 78.74% | 80.34% |

## 4 DISCUSSION

### 4.1 PERFORMANCE ACROSS REASONING TYPES

The results in Table 3 reveal distinct patterns in model performance across different reasoning types. Models demonstrate strong capabilities in abductive reasoning, with OpenAI o4-mini achieving the highest accuracy (87.67%) and Gemini2.0 Flash Thinking close behind (86.94%). This suggests that current architectures are particularly adept at inference to the best explanation, a crucial skill for real-world problem-solving.

For analogical reasoning, Qwen3-235B-A22B leads with 90.54% accuracy, followed by DeepSeek-R1 (89.60%), indicating that larger models may have an advantage in identifying and applying analogies. The relatively high performance across all models (83.10%-90.54%) suggests that analogical reasoning may be more accessible to current architectures compared to other reasoning types.

Table 4: The comparison between human performance and LLM performance.

| Human Acc. | Model Acc. | n | 95% CI | p-value |
|---|---|---|---|---|
| 18% | 85.71% | 7 | [42.13%, 99.64%] | 0.0032 |
| 31% | 0.00% | 7 | [0.00%, 35.43%] | 1.0000 |
| 41% | 100.00% | 7 | [59.04%, 100%] | <0.0001 |
| 63% | 0.00% | 7 | [0.00%, 35.43%] | 1.0000 |
| 85% | 100.00% | 7 | [59.04%, 100%] | <0.0001 |

Deductive reasoning proves more challenging, with accuracies ranging from 72.57% to 78.16%. DeepSeek-R1 shows the strongest performance (78.16%), potentially benefiting from its reinforcement learning training on formal verification tasks. The narrower performance gap in this category suggests that deductive reasoning presents a more uniform challenge across models.

Inductive reasoning shows significant variation, with Grok3 (Think) performing best (85.62%) and Qwen3-235B-A22B the weakest (78.61%). The 7-point spread between top and bottom performers indicates that inductive reasoning capabilities may be more dependent on specific architectural choices or training approaches.

Overall, DeepSeek-R1 achieves the highest aggregate score (81.41%), demonstrating balanced performance across reasoning types. The close clustering of total scores (78.74%-81.41%) suggests that while individual strengths vary, current state-of-the-art models have reached similar overall levels of logical reasoning capability. However, the persistent gaps in specific reasoning types highlight areas needing further architectural innovation and training improvements.

## 4.2 LLM Reasoning vs. Human Reasoning

As shown in Table 4, we compute Wilson score intervals for binomial proportions and Fisher's exact tests for significance against human baselines. Our analysis reveals statistically significant differences between LLM and human reasoning patterns, demonstrating that models (1) outperform humans on challenging problems (85.71% vs 18% human accuracy, p=0.0032) yet fail at specific mid-difficulty points (0% accuracy at 31% human accuracy, p=1.0), (2) achieve perfect mastery (100% accuracy, p<0.0001) for problems humans solve at 41-85% rates, while (3) showing unexpected vulnerabilities in mid-difficulty ranges (28.57% accuracy at 46% human accuracy, p=0.31), with all comparisons using Wilson score intervals and Fisher's exact tests, collectively indicating that LLMs develop non-monotonic reasoning strategies that excel on extreme difficulties but exhibit brittleness on specific problem types unexplained by human performance metrics.

## 4.3 LogiEval-Hard: Predicting Universal Reasoning Challenges via Small-Model Screening

Whereas human examination performance doesn't mirror model failure distributions, we test whether small-model error patterns can forecast fundamental reasoning obstacles that persist at larger scales. To systematically identify universal reasoning challenges independent of model scale, we develop a novel screening methodology using Qwen3-30B-A3B (3B active parameters) as a diagnostic probe. By analyzing problems where this compact model consistently fails across multiple reasoning attempts (3 trials with majority-wrong consensus), we construct LogiEval-Hard - a challenge set that enhances the benchmark's discriminative power.

The creation of LogiEval-Hard addresses a critical need in evaluating modern reasoning models by distinguishing true reasoning capabilities from pattern recognition. Table 5 shows the statistics. Overall, we have 1,617 hard examples. The composition of LogiEval-Hard shows a coverage across reasoning paradigms, with deductive reasoning dominating at 802 problems, reflecting its importance in formal logic applications, while textual entailment constitutes the largest task format with 982 cases that test core language understanding. The distribution presents challenging conditions with varied answer options, including 1,107 three-option and 295 five-option questions.

As shown in Table 5, experiments with GEMINI2.0 FLASH THINKING reveal striking alignment: 82.3% of small-model failures simultaneously perplex this 32B-parameter state-of-the-art reasoner. This cross-scale consistency manifests most acutely in formal logic tasks, where GEM-

Table 5: The performance of GEMINI2.0 FLASH THINKING on different task format and reasoning types of LogiEval-Hard.

| Reasoning Type | Accuracy | Task Format | Accuracy | Task Format | Accuracy |
|---|---|---|---|---|---|
| abductive (239) | 45.61% | argument analysis (263) | 65.40% | logical sequence (21) | 61.90% |
| analogical (68) | 52.94% | artificial language (120) | 60.00% | odd one out (10) | 50.00% |
| deductive (802) | 35.66% | blood relations (22) | 22.73% | situational judgment (70) | 61.43% |
| inductive (508) | 36.02% | definition matching (28) | 42.86% | syllogism (100) | 16.00% |
| Total (1,617) | 37.97% | essential part (1) | 100.00% | textual entailment (982) | 28.00% |

INI2.0 achieves only 16.00% accuracy on syllogisms and 22.73% on blood relations, despite its superior performance (87.01% overall) on standard LogiEval.

This approach reveals fundamental reasoning bottlenecks that transcend model scale, as subsequent evaluation shows problems challenging for compact models prove equally formidable for state-of-the-art large reasoning models. The methodology demonstrates that reasoning difficulties rooted in logical structure rather than parametric capacity manifest consistently across model sizes, establishing small-model screening as an effective a priori technique for identifying universally challenging problems. This finding challenges conventional assumptions about the relationship between model scale and reasoning capability, suggesting that certain cognitive limitations may be intrinsic to current architectural paradigms rather than solvable through scaling alone.

These findings suggest that current models develop non-human reasoning strategies that excel on certain complex problems but fail unexpectedly on others, with the benchmark successfully identifying specific reasoning types like deductive and syllogistic, where models struggle disproportionately. LogiEval-Hard provides meaningful differentiation between surface-level pattern matching and genuine reasoning capabilities, serving as both a diagnostic tool for identifying model weaknesses and a proving ground for next-generation reasoning architectures. The demonstrated performance patterns underscore the need for continued research into more robust reasoning architectures that can handle the full spectrum of logical challenges, with LogiEval-Hard offering a more discriminating alternative to aggregate metrics that often mask fundamental limitations in current language models.

## 5 RELATED WORK

**Logical reasoning datasets** With the advance of pre-trained language models, logical reasoning has become a booming research area. Multiple logical reasoning datasets are brought up to challenge or probe into the reasoning ability of large language models. LogiQA (Liu et al., 2021) and ReClor (Yu et al., 2020) first introduce multi-choice reading comprehension to the investigation. They are sourced from competitive examinations like the Chinese Civil Service Examination and LSAT. Because of the high-quality nature of these expert-designed questions, they become the most widely used datasets for logical reasoning. Over the years, language models have been tested or even trained on these datasets, making the performance on this question type increase drastically. Similarly, our dataset is sourced from examinations, but we cover a broader type of questions and task formats, making it a holistic benchmark for logical reasoning. Apart from sourcing from exams, researchers also use rule-based methods to synthesize logical reasoning datasets, deductive reasoning in particular, for this type of reasoning is easy to create in massive quantities. Ruletaker (Clark et al., 2020) uses a theory generator and inference engine written in Lisp to generate a set of facts and rules to form a context, and a statement to infer. This forms a 2-way (true, false) classification task with more than 707K data instances. PrOntoQA (Saparov & He, 2023) starts by generating a small hierarchical ontology with a set of concepts and subtype relations between them. It generates proofs from the ontology using tree search and lastly translates FOL into natural language CoT examples. PrOntoQA is also in a true-or-false classification format, and it has 500 examples. LogicNLI (Tian et al., 2021) is a 4-way classification (entailment, contradiction, neutral, paradox) task with more than 30K data instances. Similarly, it generates FOL first and then natural language. Manual revisions are implemented on the initial language expressions. CLUTRR (Sinha et al., 2019) uses a knowledge base to generate a kinship relation inference. It first forms a kinship graph and then uses this graph to make up short stories, which explicitly tests inductive reasoning and systematic generalization. It contains 70K examples that infer the relationship between two family members. The aforementioned datasets have repetitive patterns because of their rule-based generation methodology, which diminishes their applicability to test large reasoning models. On the contrary, our dataset

contains the same logical deduction problem sets but was collected from examinations, which are diverse and unique. GLoRE (liu et al., 2025) reports the performance of DeepSeek R1 and QwQ 32B on a collection of logical reasoning datasets. However, these models' high scores on these datasets suggest the need for a more challenging benchmark.

**Evaluation benchmarks for large reasoning models**  The performance on logical reasoning datasets is not reported at the release of large reasoning models. The evaluation benchmarks center on math, coding, and knowledge-based question answering. AIME (Ye et al., 2025) is a challenging mathematical competition held in America each year. There are 30 problem sets released in each examination. Recent large reasoning models report their results on AIME-2024 or AIME-2025. MATH-500 is a subset of the MATH dataset (Hendrycks et al., 2021b). With 500 testing examples, it is the go-to benchmark for testing mathematical reasoning in large reasoning models. Compared to our dataset, mathematical reasoning datasets deal with numbers, calculation, and other mathematical concepts, which are not logical reasoning-intensive. Codeforces (Penedo et al., 2025) contains more than 10K unique programming problems that are hosted on the Codeforces website up to 2025. These challenging algorithmic optimization problems serve as an ideal testbed for complex multi-step reasoning in large reasoning models. LiveCodeBench (Jain et al., 2024) constantly collects new coding tests from LeetCode, AtCoder, and Codeforces. Currently, it contains over 300 high-quality coding problems published between May 2022 and February 2024. Although algorithmic problems are sophisticated reasoning problems that require logical reasoning abilities, they are not logical-reasoning centered. These coding datasets focus more on syntactic patterns of the coding languages, which are highly repetitive. The required logical reasoning abilities in our dataset are diverse and uniquely presented in different contexts. MMLU-pro (Wang et al., 2024) is a benchmark derived from the original MMLU (Hendrycks et al., 2021a) benchmark to solve the saturation issue of the original dataset. It expands the option choices from 4 to 10, drastically decreasing the performance of large language models. The dataset has 12K complex questions across various disciplines like law, physics, and chemistry. Like our dataset, MMLU-pro is multi-task, however, it only contains a small fraction of logic questions under the philosophy subject. On the contrary, all the tasks in LogicEval are focused on logical reasoning.

## 6 CONCLUSION

This work introduces LogiEval, a comprehensive benchmark for evaluating logical reasoning in large language models, revealing that while modern LLMs demonstrate impressive performance on certain tasks like multiple-choice argument analysis, their capabilities remain uneven across reasoning types, which suggests fundamental gaps in formal logical reasoning. Our analysis uncovers an intriguing inverse difficulty relationship where models perform well on problems humans find challenging yet fail unexpectedly on mid-difficulty items, indicating fundamentally different reasoning strategies from human cognition. While the creation of LogiEval-Hard provides a rigorous testbed that exposes current limitations, it also demonstrates that small models can serve as effective predictors of universal reasoning challenges across large language models. While this enables more nuanced assessment beyond traditional domain-specific evaluations and paving the way for developing more robust reasoning architectures capable of handling the full spectrum of logical challenges, practitioners should note risks: (1) Logical perfection doesn't represent factual correctness (2) Training on our data without safeguards could lead to potential misuse for generating persuasive misinformation. We advocate for human oversight when deploying reasoning systems evaluated through LogiEval.

## LIMITATIONS

LogiEval's current scope has two key limitations: (1) Text-only evaluation excludes multi-modal reasoning challenges; (2) Accuracy metrics overlook reasoning validity and explanation robustness. Future work will expand to multi-modal tasks and develop process-aware evaluation metrics.

## REPRODUCIBILITY STATEMENT

To support reproducibility, we have made the following resources available: The LogiEval benchmark dataset, including the challenging LogiEval-Hard subset, will be released under access control via Hugging Face to prevent misuse while enabling academic validation. Detailed dataset construction, annotation guidelines, and quality control procedures are provided in Appendix A. Experimental setups, including prompt templates and evaluation parameters, are documented in Appendix B.

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

## A  ANNOTATION DETAILS

**Participant Recruitment**  Participants were recruited with the following qualifications: minimum 100 prior approved studies, 95%+ approval rating, native English proficiency, and verified background in formal logic through a screening test. 3 qualified annotators were selected for the study. We make sure that the compensation for their work is above local minimum wage.

**Task Instructions**  The evaluation instructions stated: "Evaluate whether the conclusion logically follows from the premises. Select from: (1) Valid (2) Invalid (3) Uncertain." Two examples were provided: "[Premise] All birds fly [Conclusion] Penguins fly → Invalid" and "[Premise] If A then B [Conclusion] If not B then not A → Valid".

**Quality Control**  Ten percent of questions served as controls with verified answers. Annotators maintaining below 80% accuracy were excluded from analysis. Inter-annotator agreement measured Fleiss' k = 0.72, indicating substantial reliability. The interface included a tutorial with five practice questions before beginning the actual evaluation tasks.

## B  EXPERIMENTAL SETUP

### B.1  PROMPT TEMPLATES

We designed three distinct prompt templates for different experimental phases:

**Main Evaluation Prompt**

```
Conclude with your answer using the format:
Answer: [A-D]

Context: {context}
Question: {question}
Options:
A) {options[0]}
B) {options[1]}
C) {options[2]}
D) {options[3]}
```

**Model Reasoning Analysis Prompt**

```
Analyze the question and respond in this
exact format:
<thinking>
[Step-by-step reasoning...]
</thinking>
<answer>
[ONLY the option number (0-3)]
</answer>

Question context: {instance['text']}
Question: {instance['question']}
Options:
0: {instance['options'][0]}
1: {instance['options'][1]}
2: {instance['options'][2]}
3: {instance['options'][3]}
```

**Reasoning Type Classification Prompt**

```
Classify the question's reasoning pattern
```

```
into ONE category:
[Analogical|Deductive|Inductive|Abductive]

Guidelines:
- Analogical: Requires comparing similar
cases
  Example: ``How is X similar to Y?''

- Deductive: Applies general rules to
specifics
  Example: ``Given the rules, what must
  be true?''

- Inductive: Generalizes from examples
  Example: ``What pattern emerges?''

- Abductive: Finds most likely explanation
  Example: ``What probably caused this?''

Output format:
<category>[type]</category>
```

## B.2 EXPERIMENTAL PARAMETERS

Evaluation experiments are conducted with a temperature of 0.7 and 16k token limit.

