# OpenReview forum: "Evaluating the Logical Reasoning Abilities of Large Reasoning Models"
_ICLR.cc/2026/Conference — Submitted to ICLR 2026_

### Official Review · Reviewer_KkTK · 2025-10-31

**Soundness:** 3
**Presentation:** 2
**Contribution:** 2
**Rating:** 4
**Confidence:** 2

**Summary:**

The paper introduces LogiEval, a benchmark for evaluating logical reasoning in large reasoning models (LRMs). Unlike existing benchmarks that focus on domain-specific tasks, LogiEval isolates domain-agnostic logical reasoning by curating 6,235 high-quality questions from human-administered exams. It spans four reasoning types—deductive, inductive, abductive, and analogical—and ten task formats, including syllogisms, artificial language, and situational judgment.
The authors evaluate seven state-of-the-art reasoning models (e.g., DeepSeek-R1, Claude-3.7 Sonnet, o4-mini) and find that:
•	Models excel at 4-choice argument analysis and analogical reasoning (sometimes surpassing humans).
•	They struggle with formal deductive tasks like syllogisms and blood relations.
•	Human and model difficulty patterns do not align: models often solve “hard” human problems but fail on “medium” ones.

**Strengths:**

1.	Unifying four reasoning types and ten task formats in a single, exam-sourced benchmark.
2.	Preserves original language (English/Chinese), maintaining linguistic nuance and avoiding translation bias.
3.	Evaluation of 7 top 2025 LLMs with consistent prompting and answer extraction. Includes human performance baselines and statistical significance testing.

**Weaknesses:**

1.	Most items are multiple-choice; open-ended or proof-based reasoning is underrepresented.
2.	Human accuracy is derived from historical exam pass rates, which may not reflect controlled, per-item performance.
3.	Does not assess reasoning validity or explanation quality—only final answer correctness.

**Questions:**

Please refer to the weaknesses

---

### Official Review · Reviewer_EJEJ · 2025-11-01

**Soundness:** 3
**Presentation:** 3
**Contribution:** 2
**Rating:** 2
**Confidence:** 4

**Summary:**

This paper introduces LogiEval, a benchmark for evaluating logical reasoning in large language models. The benchmark comprises 6,235 questions sourced from high-stakes examinations (LSAT, GMAT, Civil Service Exams), covering four reasoning types (deductive, inductive, analogical, abductive) and ten task formats. The authors evaluate seven state-of-the-art reasoning models released in 2025, revealing uneven performance across tasks despite strong aggregate scores.

**Strengths:**

- Clear motivation addressing the gap between domain-specific benchmarks and fundamental reasoning evaluation
- The paper is well-organzied and clearly written.

**Weaknesses:**

- Insufficient human performance analysis.
- Limited case study. It is important to eval the benchmark via case studies.
- What is the major difference between this work and previous logical evaluation benchmarks, such as GLoRE?

**Questions:**

Please refer to "Weaknesses".

---

### Official Review · Reviewer_pnNX · 2025-11-01

**Soundness:** 2
**Presentation:** 2
**Contribution:** 2
**Rating:** 2
**Confidence:** 5

**Summary:**

This paper propose a benchmark called LogicEval which evaluates the reasoning abilities of LLMs.
It is collected from several human examination, including LSAT, GMAT, Civil Service Exams.
Also a subset of LogicEval called LogicEval-hard is presented by filtering too easy examples through the results of some weak LLMs.

**Strengths:**

This paper is well-organized and easy to follow.

**Weaknesses:**

1. **Lack of Novelty**. There are too many benchmarks for evaluating logical reasoning, e.g. [1][2], even mutimodal one [3].

[1] JustLogic: A Comprehensive Benchmark for Evaluating Deductive Reasoning in Large Language Models https://arxiv.org/abs/2510.18855

[2] LogicGame: Benchmarking Rule‑Based Reasoning Abilities of Large Language Models https://arxiv.org/abs/2408.15778

[3] MME-Reasoning: A Comprehensive Benchmark for Logical Reasoning in MLLMs https://arxiv.org/abs/2505.21327

I believe it can be easy to find more with tools like DeepResearch and human double checking.

2. **Almost saturated with evaluation on out-of-dated LLMs**. Even LLMs like Deepseek-R1 / Gemini 2.0 / Claude 3.7 / Grok 3 can get 75%+ on this benchmarks as reported. But today we have:

Deepseek-R1 -> Deepseek 3.2

Gemini 2.0 -> Gemini 2.5

Claude 3.7 -> Claude 4.5

Grok 3 -> Grok 4

I believe this benchmark is almost saturated.

As the LogicEval-Hard one, please present the results on the latest LLMs.

3. **Data leakage**. There's NO NEW DATA in this benchmark, just collected existing exams, and it could be suffered from data leakage.

**Questions:**

1. Please clarity why the community still needs this benchmark at this time.

2. Please evaluate on the latest and powerful LLMs, and present the results to the readers.

---

### Meta-Review · Area_Chair_or9d · 2025-12-24

**Summary:**

This paper establishes a holistic benchmark to evaluate the logical reasoning capabilities of large reasoning models. It includes different reasoning types and task formats. The evaluation results demonstrate the strengths and weaknesses of modern reasoning models in solving different types of problems.

All the three reviewers give negative scores. Their main concerns include (1) lack of novelty, compared to existing logical reasoning works and benchmarks; (2) missing evaluations on latest models; (3) unbalanced reasoning type distributions in the benchmark; (4) limited experimental measurement and analysis.

Those raised points are critical and hard to address. The authors did not provide the responses to them. Based on those comments, AC recommended rejection.

**Reviewer Concerns:**

The authors did not provide a rebuttal.

**Reviewer Scores:**

The authors did not provide the rebuttal, so I think reviewers will not adjust their scores.

---

### Decision · Program_Chairs · 2026-01-26

Reject